# Secondary Metabolites of *Pseudomonas aeruginosa* LV Strain Decrease Asian Soybean Rust Severity in Experimentally Infected Plants

**DOI:** 10.3390/plants10081495

**Published:** 2021-07-21

**Authors:** Igor Matheus Oliveira dos Santos, Valéria Yukari Abe, Kenia de Carvalho, André Riedi Barazetti, Ane Stéfano Simionato, Guilherme E. de Almeida Pega, Sergio Henrique Matis, Barbara Gionco Cano, Martha Viviana Torres Cely, Francismar Correa Marcelino-Guimarães, Andreas Lazaros Chryssafidis, Galdino Andrade

**Affiliations:** 1Microbial Ecology Laboratory, Department of Microbiology, State University of Londrina, Londrina 86057-970, PR, Brazil; e-igor11santos@hotmail.com (I.M.O.d.S.); andrerbarazetti@gmail.com (A.R.B.); anessimionato@gmail.com (A.S.S.); guilhermeedi@uel.br (G.E.d.A.P.); sergio.henrique@uel.br (S.H.M.); barbaragionco@gmail.com (B.G.C.); 2Laboratory of Plant Biotechnology and Bioinformatics, Embrapa Soja, Londrina 86057-970, PR, Brazil; valeria.abe@gmail.com (V.Y.A.); keniadecarvalho@gmail.com (K.d.C.); francismar.marcelino@embrapa.br (F.C.M.-G.); 3Agricultural and Environmental Sciences Institute, Federal University of Mato Grosso, Sinop 78550-728, MT, Brazil; vivianatorrescely95@gmail.com; 4Agroveterinary Sciences Center, Santa Catarina State University, Lages 88501-901, SC, Brazil; andreas.107@gmail.com

**Keywords:** bioactive compounds, elicitors, *glycine max*, *Phakopsora pachyrhizi*, rust control, ASR

## Abstract

Asian Soybean Rust (ASR), a disease caused by *Phakopsora pachyrhizi*, causing yield losses up to 90%. The control is based on the fungicides which may generate resistant fungi. The activation of the plant defense system, should help on ASR control. In this study, secondary metabolites of *Pseudomonas aeruginosa* LV strain were applied on spore germination and the expression of defense genes in infected soybean plants. The F4A fraction and the pure metabolites were used. In vitro, 10 µg mL^−1^ of F4A reduced spore germination by 54%, while 100 µg mL^−1^ completely inhibited. Overexpression of phenylalanine ammonia lyase (PAL), *O*-methyltransferase (OMT) and pathogenesis related protein-2 (PR-2; glucanases) defense-related genes were detected 24 and 72 h after soybean sprouts were sprayed with an organocopper antimicrobial compound (OAC). Under greenhouse conditions, the best control was observed in plants treated with 60 µg mL^−1^ of PCA, which reduced ASR severity and lesion frequency by 75% and 43%, respectively. Plants sprayed with 2 and 20 µg mL^−1^ of F4A also decreased severity (41%) and lesion frequency (32%). The significant reduction in spore germination ASR in plant suggested that the strain of these metabolites are effective against *P. pachyrhizi*, and they can be used for ASR control.

## 1. Introduction

Asian soybean rust (ASR), caused by *Phakopsora pachyrhizi* Syd and P. Syd, is the most important disease that affects soybean crops [1] and, in severe outbreaks, it can cause production losses of up to 90% [2]. ASR is considered a threat to global food security, according to the American Association for the Advancement of Science [3]. *P. pachyrhizi* infects 93 legume species of 42 different genera [4], and it presents great genetic variability, hindering the development of resistant plants [5]. The control of ASR is mainly based on the application of chemical fungicides, with an estimated cost of around 2.2 billion dollars per year for Brazilian farmers [2]. In addition, the emergence of antifungal resistant strains decreases fungicide efficiency [6]; thus, new strategies and molecules for ASR control are urgently needed.

Plants are under constant stress, using structural and chemical defense mechanisms to survive biological, chemical and environmental agents. There are different mechanisms involved in the activation of their defense system, such as the hypersensitive responses [7] and pathogenesis-related (PR) proteins [8]. The PR proteins are effectively induced by harmful stimuli, activating the systemic acquired resistance (SAR) [9]. They are classified into 17 families (PR-1 to PR-17), according to their biochemical and biomolecular properties [10]. 

The synthesis and storage of PR proteins is crucial for the plant self-defense mechanism against pathogenic infections and abiotic stress [11,12]. For instance, PR2 and PR3 are generated in large quantities when plants respond to injuries or infection by fungal, bacterial or viral pathogens [13,14]. 

Although it is known that most PR proteins have antifungal properties, their active molecular mechanisms are not well understood, except for PR-2 (glucanases) and PR3 (chitinases) [15]. The PR expression is regulated by many signaling compounds, such as abscisic acid (ABA), ethylene (ET), jasmonic acid (JA), and salicylic acid (SA) [16]. The genetic expression of different proteins can be used as a bioindicator of SAR activation.

The phenylpropanoid pathway is an important secondary metabolism route and produces a wide variety of metabolites [17]. Phenylalanine ammonia-lyase (PAL) is involved in the formation of lignin, suberin, phytoalexins, stilbenes, coumarins and other flavonoids [18]. The biosynthesis of glyceollins occurs via the phenylpropanoid pathway in soybean [19]. Glyceollins are soybean-derived phytoalexins that accumulate in the seeds in response to abiotic stress conditions [20]. Glyceollins inhibit many phytopathogen species, especially fungi [21], and their concentration increased when soybean leaves were infected with *Phakopsora pachyrhizi* [22].

The biosynthesis of lignin, as well as the plant resistance and stress tolerance, are dependent on *O*-methylation. *O*-methyltransferases (OMTs) are a large family of plant enzymes that bind methyl groups to the oxygen of many secondary metabolites [23]. The caffeoyl CoA OMTs (CCoA OMTs) act on phenolic hydroxyl groups of hydroxycinnamoyl CoA esters, while the carboxylic acid OMTs change aliphatic carboxyl groups, and other OMTs methylate other metabolites, such as hydroxycinnamic acids, flavonoids, and alkaloids [23]. In soybean plants, different OMTs were described [24].

The SAR activation begins with chemical signals at the induction/infection site, which spreads throughout the plant organism, preparing their tissues for faster and more efficient responses against pathogens [25]. Salicylic acid levels and the biosynthesis of PR proteins are increased by SAR, immunizing the entire plant against future infections [26]. Many chemical compounds are known as SAR inducers, such as acibenzolar-S-methyl (ASM), 2.6-dichloroisonicotinic acid (INA) and beta-aminobutyric acid (BABA), but they are not always effective alone.

Beneficial bacteria are great producers of compounds that induce systemic resistance in plants [27,28], especially the genus *Pseudomonas* [29,30]. The application of F4A, a cell free-fraction of *P. aeruginosa* LV strain culture, induced resistance in tomato plants against *Pectobacterium carotovorum*, increasing the biosynthesis of defense-related enzymes, such as peroxidases and phenylalanine ammonia-lyase (PAL) [31]. 

The F4A fraction contains four important metabolites: phenazine-1-carboxylic acid (PCA), phenazine-carboxamide (PCN), indol-3-one (IND) and an organocopper antimicrobial compound (OAC), and when applied on citrus plants, induced the expression of genes encoding PR-2 protein, one of the markers of SAR [32] reducing the infection of *Candidatus*
*liberibacter* asiaticus [33]. 

Some compounds can activate soybean defense mechanisms, reducing damages caused by ASR. The foliar application of saccharin induced SAR and reduced ASR severity [34]. 

Acibenzolar-S-Methyl (ASM) sprayed on soybean leaves, and the addition of calcium silicate to the soil, stimulated defense enzymes and decreased the severity of ASR [35]. On the other hand, other studies found that ASM was not effective in the control of ASR under field conditions [36,37].

The objective of the present study was to evaluate the effects of secondary metabolites produced by *P. aeruginosa* LV strain applied on the *P. pachyrhizi*–soybean pathosystem. For this purpose, the following hypotheses were tested: (1) *P. aeruginosa* LV strain metabolites upregulates the expression of resistance related genes in soybean; (2) the direct application of these metabolites may help to control infection by *P. pachyrhizi* by in vitro and in vivo experiments.

## 2. Results

### 2.1. The Evaluation of F4A Fraction on P. pachyrhizi Spore Germination

The germination rate of non-treated spores was of 88%. The dose/effect of F4A concentration showed high antifungal activity on spore germination of *P. pachyrhizi*. F4A at the concentration of 10 µg mL^−1^ reduced spore germination by around 50%, while 100 µg mL^−1^ completely inhibited spore germination (Figure 1 and Figure 2).

### 2.2. Gene Expression of Soybean Plant Treated with Secondary Metabolites of P. aeruginosa LV Strain

The qRT-PCR analysis showed that only OAC induced the upregulated expression of PR-2, PAL and OMT defense-related genes (Figure 3). The PR 2 gene expression level increased eight times 24 h after treatment, when compared with the controls. The expression levels of PAL and OMT genes increased at 24 and 72 h, with the highest expression observed at 72 h (Figure 3b). There was no upregulated expression of PR-2, PAL and OMT at 168 h (Figure 3c).

### 2.3. ASR Control in Soybean Plants Treated with Secondary Metabolites of P. aeruginosa LV Strain

The secondary metabolites caused different effects on decreasing ASR severity when applied together in the F4A fraction or separated in purified compounds. The most effective concentrations of F4A fraction were 2 and 20 µg mL^−1^ when compared to the controls (Figure 4). 

PCA at 60 µg mL^−1^ was the most effective treatment, when compared to the other concentrations of PCA, all concentrations of PCN, OAC and even to the F4A fraction. Disease severity also decreased in plants treated with 5 and 50 µg mL^−1^ of PCN and 0.5 and 5 µg mL^−1^ of OAC, but their efficiency was lower than F4A at 2 and 20 µg mL^−1^ (Figure 4).

The pattern found in lesion frequency was similar to that observed in disease severity (Figure 5). 

DMSO + mineral oil (DMSO); semi-purified fraction (F4A); phenazine-carboxylic acid (PCA); phenazine carboxamide (PCN); organocopper compound (OAC). The numbers after compound legend indicate their concentration (µg mL^−1^). Different letters indicate significant differences by Tukey test at *p* < 0.05. Bars indicate standard error. CV = 41.5%.

Plants treated with 2 and 20 µg mL^−1^ of F4A had around 32% less lesions when compared to the control. The application of 60 µg mL^−1^ of PCA reduced lesion frequency by 42% and the OAC at 0.5 and 5 µg mL^−1^ decreased the frequency by 30 and 29%, respectively. The most effective concentrations of each compound are represented in Figure 6.

## 3. Discussion

The F4A fraction was applied in *C. sinensis* cv. Valencia at a concentration of 100 µg mL^−1^, being proved a valuable tool for the control of Huanglongbing disease [33]. In the present study, the maximum concentration used was of 20 µg mL^−1^, once phytotoxic effects were observed in soybean leaves at concentrations above 20 µg mL^−1^ (data not shown).

To our best knowledge, this is the first time that PCA was tested in soybean against *P. pachyrhizi*. The PCA solution at 60 µg mL^−1^ did not eliminate ASR, but greatly reduced disease severity and lesion frequency when compared to the controls. PCA treatments at concentrations above 500 µg mL^−1^ were able to control *P. capsici* in pepper and *Colletotrichum orbiculare* in cucumber [38]. The same authors observed that the minimum inhibitory concentration of PCA for many phytopathogenic fungi was greater than 100 µg mL^−1^. However, in the present study, the PCA concentration that generated the best results against *P. pachyrhizi* was much lower. Therefore, it may be possible that higher concentrations of PCA result in better ASR control, eventually reaching the complete pathogen elimination. 

Some experiments evaluated the expression of many defense genes during *P. pachyrhizi* infection in susceptible and resistant soybeans genotypes, and PAL and OMT genes were the most important among them [39,40]. In experiments with genetic manipulation of ASR-resistant plants, it was demonstrated that PAL, OMT and PR-1 genes were involved in the resistance mechanism, as the plants became susceptible when such genes were silenced [41,42].

In the presented experiment, the OAC upregulated the expression of the defense-related genes PR-2, PAL and OMT. Phenylpropanoid pathway constitutes one of the most important plant defense pathways against pathogens, especially in soybean resistance to *P. pachyrhizi* [22,42,43]. PAL is the first enzyme of the phenylpropanoid pathway, providing precursors for OMT to produce phytoalexins, such as glyceolin and lignin [22,44]. Both PAL and OMT are expressed in ASR resistant soybean genotypes [45]. OAC induced higher expression of the main defense genes against *P. pachyrhizi*, and reduced disease severity and lesion frequency. However, PCA at 60 µg mL^−1^ had better performance against the pathogen, regarding the reduction in disease severity and lesion frequency. Possibly, other defense mechanisms not measured in the present study were involved.

ASR resistance is directly related to the intensity and timing of defense-related gene expression [45]. The ASR susceptible genotypes have a peak of defense-related genes expression 12 h after spore inoculation, followed by a second increase at 96 h, while resistant genotypes expressed these peaks at 12 and 72 h, which corresponds to the time of fungal penetration and haustoria formation [41,42]. 

Despite the strong correlation between gene expression and metabolite production, enzymatic activity and translational regulation may change dynamically, with an eventual decrease in the level of some key metabolites [46]. From 570 expressed genes and 507 proteins evaluated in an ASR-resistant-soybean genotype, only nine gene/protein correlations were found during pathogenic infection, suggesting that the production and accumulation of proteins and metabolites in *P. pachyrhizi*–soybean interaction is influenced by other biological processes, beyond gene transcription [41]. 

In the present study, high expressions of PAL, OMT and PR-2 genes were observed at 24 and 72 h after OAC application. The upregulated defense-related genes stimulated by OAC, without the complete disease control, lead to two assumptions: (1) the increased expression of the measured genes did not occur during the critical moment of fungal establishment in leaf tissue thus it was unable to block the infection of the host plant; (2) the genetic expression did not result in the production of protective phytoalexins or antimicrobial proteins by the plant. However, it is possible that spraying the plants with OAC after pathogen inoculation could promote better results for the control of ASR, by the alignment of fungal infection with the activation of plant defenses.

Only the OAC compound was able to induce the upregulation of the defense-related genes evaluated in these experiments. However, the semi-purified extract F4A and PCA presented higher efficiency in controlling the experimental *P. pachyrhizi* infection in soybean. The results found in spore germination, genetic expression, disease severity and lesion control show that *P. aeruginosa* metabolites have great potential against this utterly important fungal disease. Further studies will be carried out for optimizing formulation and concentration of each compound, as well as the application timing of the treatment, with the objective of developing the most efficient combination for ASR control under field conditions.

## 4. Materials and Methods

### 4.1. Metabolites Production

*P. aeruginosa* LV strain (GenBank: QBLE00000000.1) was isolated from a citrus canker lesion of *Citrus sinensis* cv. Valencia fruit in Astorga, Brazil [47]. The strain was cultivated in nutrient broth (NB) plus 0.01% CuCl_2_ 2H_2_O for 28 °C/10 days. A semi-purified fraction, named F4A, was extracted by liquid vacuum chromatography (LVC) from a cell-free supernatant [31]. The compounds PCA, PCN and OAC were purified by flash chromatography (FC) using a 10 cm column containing silica 0.04–0.063 mm. The PCA and OAC were extracted with 200 mL of dichloromethane plus ethyl acetate [95:5]. PCN was extracted with 200 mL of dichloromethane plus ethyl acetate [50:50]. The purity level of these compounds was checked by HPLC [33], and after that, the compounds were dissolved in dimethyl sulfoxide (DMSO) and purified water before sprayed on plants. The concentration of each compound in F4A total volume was PCA 30%, PCN 25% and OAC 25% [33].

### 4.2. The Evaluation of F4A Fraction on P. pachyrhizi Spore Germination

Aliquots of 1 mL containing 4 × 10^4^ mL^−1^ spores of *P. pachyrhizi* were distributed in Petri dishes containing 1.5% water agar and 0.5% of DMSO plus 1, 10 or 100 µg mL^−1^ of F4A in each plate, being incubated for 12 h at 25 °C. Three replicates were performed for each F4A concentration. Water agar plus DMSO was used as control. The spore germination (%) was evaluated by optical microscopy (100X).

### 4.3. Gene Expression of Soybean Plants Treated with Secondary Metabolites of P. aeruginosa LV Strain

Before this experiment, many other preliminary experiments were carried out to verify and standardize the experimental conditions, such as the concentrations of applied compounds, the higher yields of expressed genes, and other variables for better qRT-PCR analysis. According to these preliminary results (data not shown), three genes (PR-2, PAL and OMT) were chosen as bioindicators of SAR, after the plants were treated with *P. aeruginosa* LV strain secondary metabolites, as described below. The endogenous gene encoding β-actin of soybean was used as internal control.

Seeds of *Glycine max* cv. BRS 184, a highly ASR-susceptible cultivar, were pre-germinated for 72 h at 25 °C and transferred to two-liter pots filled with unsterilized soil. Plants were kept in a growth chamber (14 h/10 h photoperiod day/night, 28 ºC and 50% RH) and watered daily. The experimental design was entirely randomized, with three replicates and three sampling times per treatment. Four treatments were tested: the semi-purified fraction F4A (20 µg mL^−1^), and the pure compounds PCA (6 µg mL^−1^), PCN (5 µg mL^−1^), OAC (5 µg mL^−1^). Plants were sprayed with 5 mL per plant of each treatment in the 21st day after sowing. Plants sprayed with distilled water plus DMSO (0.5%) and mineral oil (0.25%) were used as controls. 

The second trifoliate leaf of each plant was collected at 24, 72 and 168 h after treatment. They were quickly transferred to liquid nitrogen and stored at −80 °C. These leaves were macerated in liquid nitrogen, using materials previously treated with RNase AWAY ^®^ (Merck). The RNA extraction was performed using Trizol (Invitrogen, Waltham, MA, USA) method, following manufacturer recommendations. The extracted RNA was resuspended in RNase-free water. RNA concentration of each sample was measured using a NanoDrop™ spectrophotometer (Thermo Fischer Scientific, Waltham, MA, USA) and the samples were stored at −80 °C. Samples were adjusted to the final concentration of 3 µg µL^−1^ and treated with DNase Turbo™ (Invitrogen), according to the manufacturer indications. The cDNA synthesis was performed with Superscript™ III (Invitrogen), according to the manufacturer protocol. The qRT-PCR was executed in a 7500 Real Time PCR System (Applied biosystem^®^) with 96-well capacity. The oligonucleotides used are presented in Table 1. 

### 4.4. ASR Control in Soybean Plants Treated with Secondary Metabolites of P. aeruginosa LV Strain

Preliminary experiments were carried out, testing different conditions for better evaluating the activity of secondary metabolites of *P. aeruginosa* LV strain against ASR. The final experiment was carried out at a greenhouse in Embrapa Soja-Londrina. The experiment was a split plot design with three concentrations of F4A, PCA, PCN and OAC (Table 2), with four replicates. Distilled water plus DMSO (0.25%) and mineral oil (0.25%) were used to dilute the compounds, and they were applied pure on the control plants.

Soybean seeds cv. BRS 184 were sowed in three-liter pots containing the same non-sterilized soil as described above. Pots were fertilized with the equivalent of 250 Kg ha^−1^ of N-P-K 10-10-10. The temperature was adjusted to 28 ºC and plants were watered when needed. After 20 days of sowing, plants were uniformly sprayed with each compound solution (5 mL per plant). *P. pachyrhizi* inoculation was performed by spraying a spore suspension (10^4^ spores mL^−1^) at the abaxial side of leaflets, 24 h after treatment. 

Two leaflets were collected from the second trifoliate leaf of each plant for phenotypic evaluation, 14 days after inoculation. The lesion frequency was determined according to the number of lesions observed in 1 cm^2^, on each side of the leaflet. The disease severity was evaluated as the percentage of the leaf area covered by lesions caused by the fungus, analyzed by the software programs Photoshop CS6 and Quant [48].

### 4.5. Statistical Analysis

Spore germination, severity and lesion frequency that attended normality and homogeneity according to the Shapiro–Wilk and Bartlett tests were evaluated using Tukey’s test (*p* < 0.05), in order to verify the difference of each treatment compared to control using RStudio software. For gene expression data analysis, Student’s *t*-tests were performed to compare changes in gene expression levels, using the software program 7000 System SDS (Applied biosystems^®^).

## Figures and Tables

**Figure 1 plants-10-01495-f001:**
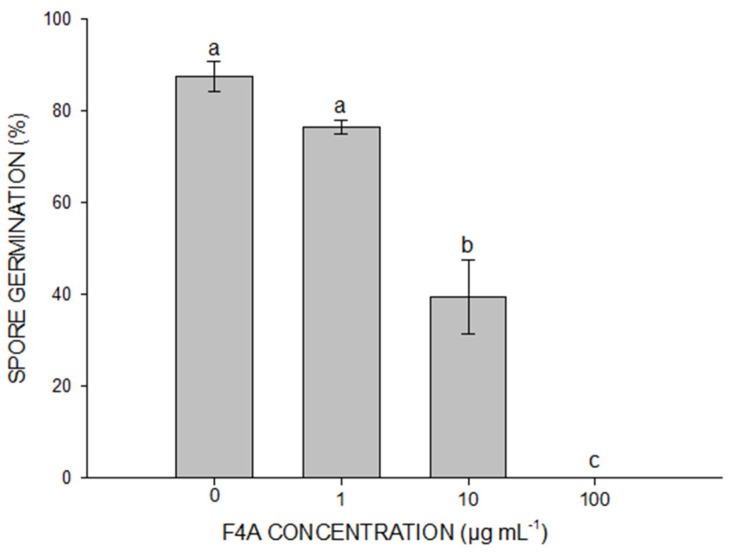
Quantification of *P. pachyrhizi* spore germination in water agar plus with different concentrations of the semi-purified fraction F4A. Different letters indicate significant differences by Tukey test at *p* < 0.05. Bars indicate standard deviation. CV = 8.6%.

**Figure 2 plants-10-01495-f002:**
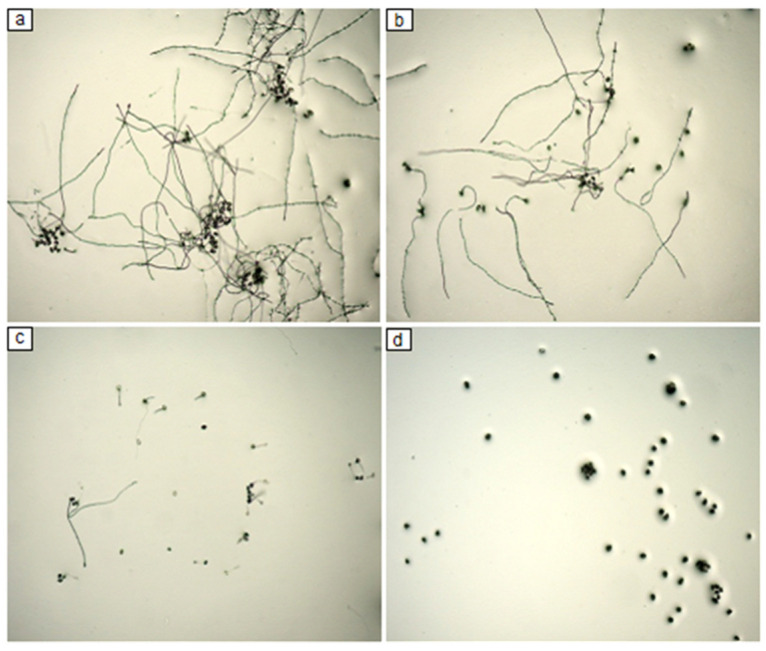
Spore germination of *P. pachyrhizi* in water agar plus with different concentrations of the semi-purified fraction F4A. (**a**) control, (**b**) 1 µg mL^−1^, (**c**) 10 µg mL^−1^ and (**d**) 100 µg mL^−1^.

**Figure 3 plants-10-01495-f003:**
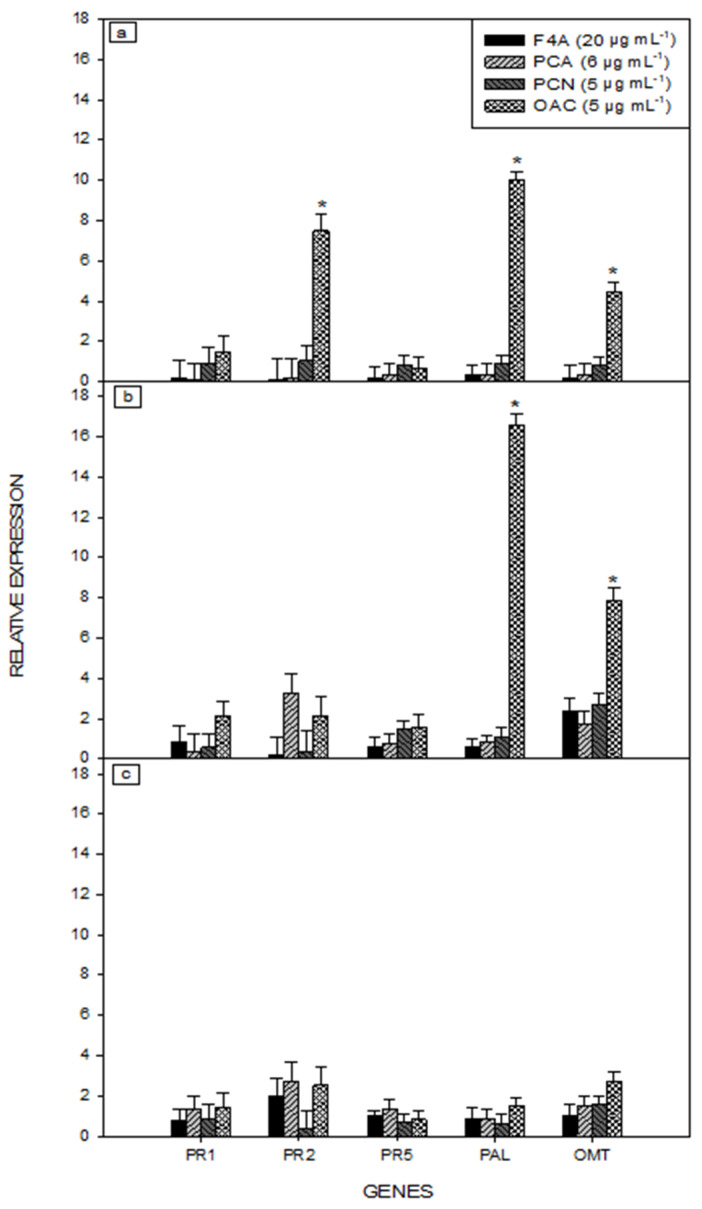
Defense-related gene expression in soybean plants treated with different secondary metabolites produced by *P. aeruginosa* LV strain. Expression of PR-1, PR-2, PR-5, phenylalanine amonia-lyase (PAL) and O-methyltransferase (OMT) at (**a**) 24 h, (**b**) 72 h and (**c**) 168 h after metabolites application. The legend corresponds to the treatments: semi-purified fraction (F4A); phenazine-carboxylic acid (PCA); phenazine carboxamide (PCN); organocopper compound (OAC). Bars indicate the standard deviation. * Significant differences, according to Student’s *t* test (*p* < 0.05).

**Figure 4 plants-10-01495-f004:**
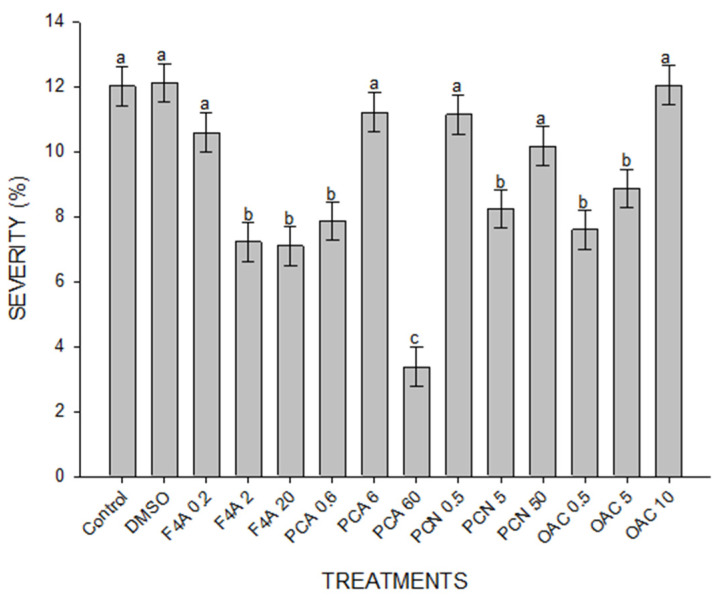
Disease severity caused by *P. pachyrhizi* in soybean leaves treated with different secondary metabolites produced by *P. aeruginosa* LV strain, 14 days after inoculation. The fungal spores were inoculated 24 h after treatment. Distilled H_2_O (Control); DMSO + mineral oil (DMSO); semi-purified fraction (F4A); phenazine-carboxylic acid (PCA); phenazine carboxamide (PCN); organocopper compound (OAC). The numbers after compound legend indicate their concentration (µg mL^−1^). Different letters indicate significant differences by Tukey test at *p* < 0.05. Bars indicate standard deviation. CV = 61.94%.

**Figure 5 plants-10-01495-f005:**
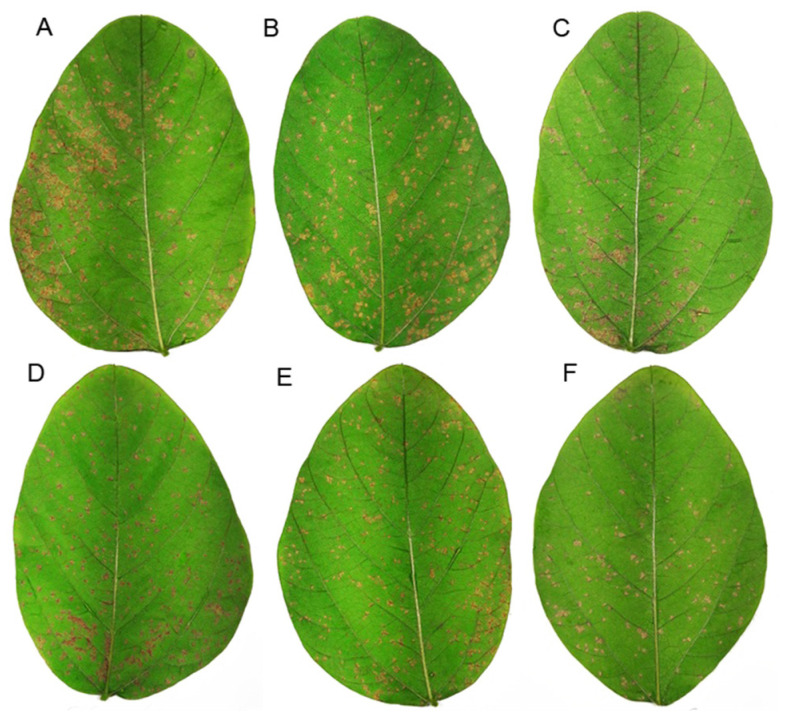
Lesions caused by *P. pachyrhizi* in soybean leaves treated with different secondary metabolites produced by *Pseudomonas aeruginosa* LV strain, 14 days after inoculation. The fungal spores were inoculated 24 h after treatment. (**A**) Distilled H_2_O; (**B**) DMSO + mineral oil; (**C**) F4A 20 µg mL^−1^; (**D**) phenazine-carboxylic acid (PCA) 60 µg mL^−1^; (**E**) phenazine carboxamide (PCN) 5 µg mL^−1^; (**F**) organocopper compound (OAC) 0.5 µg mL^−1^.

**Figure 6 plants-10-01495-f006:**
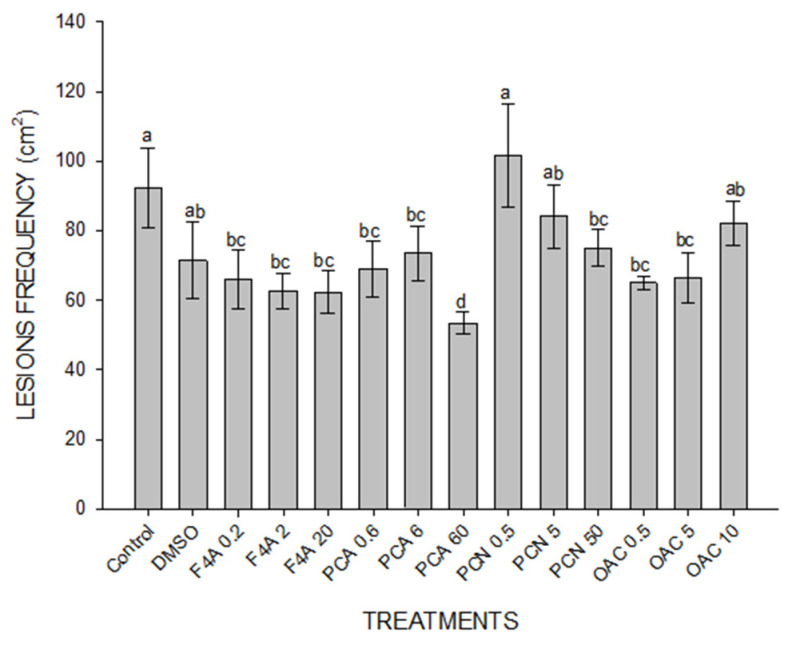
Frequency of lesions caused by *P. pachyrhizi* in soybean leaves treated with different secondary metabolites produced by *P. aeruginosa* LV strain, 14 days after inoculation. The fungal spores were inoculated 24 h after treatment. Distilled H_2_O (Control). Different letters indicate significant differences by Tukey test at *p* < 0.05. Bars indicate standard deviation. CV = 61.94%.

**Table 1 plants-10-01495-t001:** Oligonucleotides used in qPCR analyses.

Oligonucleotide	Sequence 5 ‘–3 ‘
β-actin F	GAGCTATGAATTGCCTGATGG
β-actin R	CGTTTCATGAATTCCAGTAGC
PR-1 F	AGAGGCAGAGGTGGGTTCT
PR-1 R	TCACCAACAAAGTTGCCAG
PR-2 F	TGAAATAAGGGCCACGAGTCCAAATG
PR-2 R	ATGGTACATGCAGACTTCGAATGCAGAT
PR-5 F	CTCATGCACCAGTATTCCC
PR-5 R	AAGCTTTGTAGTTGGTCC
PAL F	CAAACATCGGCAGATTACTCC
PAL R	CTGGAATGTCTTGGAGATTGG
OMT F	TGGCTAGTCACTCCATGCTATC
OMT R	AACGAGACACCATCAGCATC

Note: F4A—purified fraction; PCA—phenazine-carboxylic acid; PCN—phenazine carboxamide; OAC—organocopper compound.

**Table 2 plants-10-01495-t002:** Description of treatments used in the severity experiment, the products used and the respective concentrations.

Treatment	Product	Concentration (µg mL^−1^)
1	H_2_O	
2	H_2_O + DMSO + mineral oil	
3	F4A	0.2
4	F4A	2
5	F4A	20
6	PCA	0.6
7	PCA	6
8	PCA	60
9	PCN	0.5
10	PCN	5
11	PCN	50
12	OAC	0.5
13	OAC	5
14	OAC	10

Note: F4A—purified fraction; PCA—phenazine-carboxylic acid; PCN—phenazine carboxamide; OAC—organocopper compound.

## Data Availability

The datasets generated during and/or analyzed during the current study are available from the corresponding author on reasonable request.

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
