# Peer review of "Secondary Metabolites of Pseudomonas aeruginosa LV Strain Decrease Asian Soybean Rust Severity in Experimentally Infected Plants"

_plants, 2021, doi:10.3390/plants10081495_

Round 1

Reviewer 1 Report

The study evaluates the effects of secondary metabolites produced by Pseudomonas aeruginosa LV strain applied on the Phakopsora pachyrhizi-soybean pathosystem. The topic of paper is interesting and in the aim of the journal. The paper requires minor revision because needs some additional details to be considered for publication.

Specific comments and suggestions for improving the paper are:

Line 20, please, specify  PAL, OMT and PR2

Introduction, there are too many line breaks, please check…

Line 54-55, please, write out ABA, ET, JA, and SA

Lines 112-113, please, delete (a) control, (b) 1 μg mL-1, (c) 10 μg 112 mL-1 and (d) 100 μg mL-1.

Line 256, please, write out PCA, PCN and OAC

Line 260, please, specify the HPLC brand and the operating conditions.

Line 262, please, check “total volume was PCA 30%, PCN 25% and OAC 25%” the total is not 100.

Fig 5, please check caption…  add the corresponding A,B,C,D of the figure

Line 300, table 1 is missing!

Line 307, table 2 is missing!

Author Response

Dear Reviewers

We accept all changes that you sugested, and we also sending a tracked version of our manuscript.

Reviewer 2 Report

The manuscript submitted by Oliveira et al. describes the impact of secondary metabolites isolated from P. aeruginosa on the control of Phakopsora pachyrhizi. The results obtained are very good and interesting, in addition to the fact that the work is complete and provides important information for the scientific community.

General comments:

In my opinion, it would be good if agronomic variables of the crop were included in which the severity of the pathogen is evaluated. This would further complement the work presented.

Specific comments:

Check the abstract, as it presents several formatting errors, perhaps derived from the conversion to pdf.

Check the use of subscripts and superscripts throughout the entire manuscript.

Figure 1. If the germination of the spores is zero, it should not be considered in the statistical analysis. Instead it could be "undetected".

If possible, the quality of the Figures should be improved.

Author Response

We accept all changes that you sugest.

Reviewer 3 Report

The text is well written and the research design is appropriatee.

Line 125: Please change "The PR 2 gene" to "The PR-2 gene".

Author Response

We accept all changes that you sugest.
